# Analysis of the Severity and Cause and Effect of Occupational Accidents in South Korea

Jérémie Tuganishuri [1,†], Sang-Guk Yum [1,†], Manik Das Adhikari [1] and Tae-Keun Oh [2,*]

1   Department of Civil and Environmental Engineering, Gangneung-Wonju National University, Gangneung 25457, Republic of Korea; tuganishuri.jeremie@gwnu.ac.kr (J.T.); skyeom0401@gwnu.ac.kr (S.-G.Y.); manik@gwnu.ac.kr (M.D.A.)
2   Department of Safety Engineering, Incheon National University, Incheon 22012, Republic of Korea
*   Correspondence: tkoh@inu.ac.kr; Tel.: +82-032-835-829
†   These authors contributed equally to this work.

**Abstract:** The industrial sector in South Korea has recently undergone significant growth; however, it is also known for its hazardous workplaces. Occupational accidents have had a widespread impact across various industries; therefore, the identification of accident-influencing factors is crucial to improve workplace safety. We analyzed the occupational accident database from the Ministry of Economy and Finance to examine the influencing factors, including worker information, project details, time-related variables, and accident descriptions. Exploratory and correspondence data analyses were performed to identify patterns and relationships between variables. We applied multinomial logistic models and random forest algorithms to understand the correlation between victim status and independent variables. Results showed that 67% of all accidents occurred among workers with less than one month of employment. The multinomial regression model achieved a prediction accuracy of 97.66% with a kappa value of 0.846, outperforming the random forest model (kappa = 0.844). The receiver operating curve illustrated that the random forest had higher misclassification rates when distinguishing between injuries and fatalities. To mitigate accidents among new workers, enhanced safety training and protective measures are needed to enforce a healthy workplace. This study contributes to ongoing efforts to advance workplace safety, reduce occupational accidents, and promote a healthier working environment.

**Keywords:** occupational accidents; healthy work environment; random forest; multinomial model; multi-class problems





## 1. Introduction

### 1.1. Background of the Study

Globally, occupational accidents claim approximately 2.3 million lives annually among workers, with an additional 340 million workers enduring injuries and around 160 million individuals being affected by work-related illnesses [1]. The consequence of these occurrences is profound, affecting the afflicted workers and their families. These incidents lead to a reduction in workdays due to injuries and the need for disease treatment, thereby impeding overall wellbeing and exerting a negative impact on the economy at large.

In Korea, the surge in industrial development witnessed in recent decades has corresponded with a notable uptick in occupational accidents, particularly within the construction sector. This was largely attributed to the inherent hazards in various work environments and the substantial demand for skilled and unskilled workers. Consequently, the occurrences of occupational accidents have surged over the past few decades, leaving a trail of physical injuries, work-related sicknesses, and, tragically, fatalities among workers. The characteristics of occupational accidents in South Korea revealed that half of all

workplace accidents between 1991 and 1994 occurred within the initial year of employment [2]. This highlights the critical role of work experience as a pivotal determinant in understanding the root causes of such accidents.

A safe workplace setting is important for workers' productivity [3–5], and a clean environment is associated with no risk of bumping into objects [6]. Additionally, working at surface level is associated with no risk of falling from height [7]. Kang [8] reported a decrease in injuries resulting from entanglement with objects, while incidents involving slips, falls on the same level, and falls from heights exhibited an increase in the construction industry in South Korea from 1970 to 2000. Accident hazards within the workplace are also different based on the working section, such as warehouse, production, and finishing areas. Additionally, accidents originate from various sources, including production materials, machinery, misplaced tools, and the behavior of both workers and supervisors. These last sources consist of workers ignoring the use of protective equipment, failing to conduct proper workplace inspections, and not ensuring the provision of necessary safety devices [9]. Further, Jeong [2] studied the location of injuries on the body and analyzed both fatal and non-fatal accidents in South Korea, emphasizing the importance of detailed information about the wounded part of the body for assessing accident severity. Similarly, Al-Abdallat et al. [10] reported that the body part, such as the head, significantly affects the severity of the accident and the level of fatality and permanent disability in Jordan. Furthermore, time-based variables can play a critical role in the occurrence of the accident and its severity. For example, nightshift workers are exposed to sleep deprivation due to working in the circadian phase naturally programmed for sleeping. This leads to poor sleep during the daytime, leading to chronic sleep disturbance, occupational accidents, and other diseases such as obesity [11]. The day of the week and month of the accident may also be associated with accidents based on the type of occupation, especially outdoor activities such as agriculture, construction, mining, etc. [12,13]. Szóstak [14] utilized the timing of accident occurrences, identifying three distinct peak points: between 7 and 8 a.m., the midday hour, and 2 to 3 p.m. Additionally, the project value is believed to be an influencing factor of accidents. Smaller projects are thought to have a higher rate of accidents because fund managers may prioritize budget maximization and disregard safety costs. On the contrary, large projects may have enough funds to finance the safety cost. The study conducted in the construction field in Spain by Pellicer et al. [15] revealed that the cost of ensuring safety measures can account for up to 5% of the total project cost. Thus, the size of the project is a crucial factor to consider in the analysis of occupational accidents.

It was observed that the influencing factors including worker information (i.e., age, work experience, gender, employment category), features of a project (i.e., project value, number of employees, project type), time-based variables (i.e., hour, day, and month of the accident), and accident description (i.e., diagnosis, wounded part, lost workdays) could play a significant role in the identification of occupational accident severity. Therefore, a comprehensive analysis of occupational accidents can be a first step in finding the actual cause of accidents to enforce preventive measures for accident mitigation, enhancing the long-term productivity and sustainability of the industries.

### 1.2. Literature Review

Due to the unstable state of the industrial safety management system, the analysis of the severity and cause–effect of occupational accidents can contribute to the improvement of workplace safety. Consequently, extensive research has been carried out in recent decades to enhance safety performance on construction sites [3,4,16]. The socio-demographic factors, including gender, age, occupation, and work experience in relation to accident occurrences, have been widely analyzed to increase construction site safety performance [9,17,18]. These factors are vital in identifying vulnerable societal groups, thus aiding the implementation of protective policies tailored to those specific groups. Notably, socio-demographic information carries greater significance for developed countries grappling with population aging challenges [19]. Furthermore, considering socio-economic variables is important in

analyzing the provision of post-accident recovery support for victims [20]. Kim and Lee [9] also highlighted substantial disparities in victim characteristics, examining variables such as industry type, business size, worker age, accident patterns, work duration, and accident year. Additionally, Jo et al. [21] analyzed the construction accidents and revealed that the elevated incidence and mortality rates among male workers were predominantly attributed to falls. Therefore, it is imperative to explore the interplay among the various occupational accident variables to assess the features of accidents and gain insights into them.

Recently, numerous researchers [16,18,22–24] applied machine learning and deep learning algorithms to unravel the cause–effect relationships of accidents, highlighting the influence of human factors, equipment malfunctions, and workplace conditions. Kim and Park [18] utilized multilinear regression to model the correlation between economic factors and accidents. Their primary focus was to examine how accidents vary in response to fluctuations in the economy. Kang and Ryu [25] used random forest methodology to predict the type of accidents. The primary objective of their study was to determine the accident type based on various influencing factors, irrespective of their severity. Moreover, Kim et al. [26] analyzed the relationship between working contracts and departments and the frequency of accidents in shipbuilding. The focus was to analyze the frequency based on stress induced by the type of work conducted in different departments and climates. Jeong [2] applied descriptive statistics to analyze the characteristics of accidents in manufacturing companies; the research aimed to illustrate the accurate influencing factors of accidents for mitigation and prevention. Rafindadi et al. [27] studied the causes and preventive measures of fatal fall-related accidents in the construction industry.

The existing studies focused on characterization, the identification of a cause, accident types, etc.; however, the severity of the accident and the influencing factors were not thoroughly analyzed. In the present study, we focus on analyzing the severity of occupational accidents with respect to their influencing factors to identify the list of the most influencing factors for accident mitigation. Here, the severity is defined as the state of the victim after the accident: injured, occupational diseases, and dead. The main objective of this study is to combine the socio-economic data, project-related information, and time-based variables to predict the severity of occupational accidents using multinomial logistic models and random forest algorithms. Consequently, this work analyzes the combined effect of worker's personal information, time-based information, working conditions, and the part of the body injured on the severity of occupational accidents.

## 2. Data and Methodology

The following steps were followed to predict the severity of accidents. First, the dataset was organized and cleaned to facilitate all analyses. Subsequently, an exploratory data analysis was conducted, utilizing various visualization techniques to identify patterns and clusters. Additionally, a multinomial regression model was developed to probabilistically examine the relationship between the severity of occupational accidents and explanatory variables. Figure 1 illustrates schematically detailed steps undertaken to perform all analyses.

### 2.1. Data Description

The dataset used in this study was collected from the Ministry of Economy and Finance of South Korea and contains occupational accident records for the year 2019. It tabulates the basic information about the accident victims, namely, gender, age, skills, work experience or simply the length of employment, and the status of the victim, which has three states: injured, diseased patient, or dead. The injury state stands for the physical wound or fracture of the victims, while the disease state represents the infection and other non-visible maladies that resulted from the accident. In addition, the wounded part of the victim, the form of occurrence that is the accident source, and the number of workdays lost due to the accident were considered variables in the modeling process. The wounded part variable was created in the dataset based on the international classification of diseases code

ICD-11 [28]. Moreover, the information about the time of the accident (i.e., month, day, hour) was considered due to the variation in strength and focus of the human body associated with the change in the time [14,29]. Furthermore, project information, namely, project value, number of employees, and project phase, were considered to examine their effect on the frequency and type of accidents. Detailed information about the considered variables in the study is summarized in Table 1. The time variable was considered categorical because the study deals with the frequency and severity of an event that happened during an interval of time.

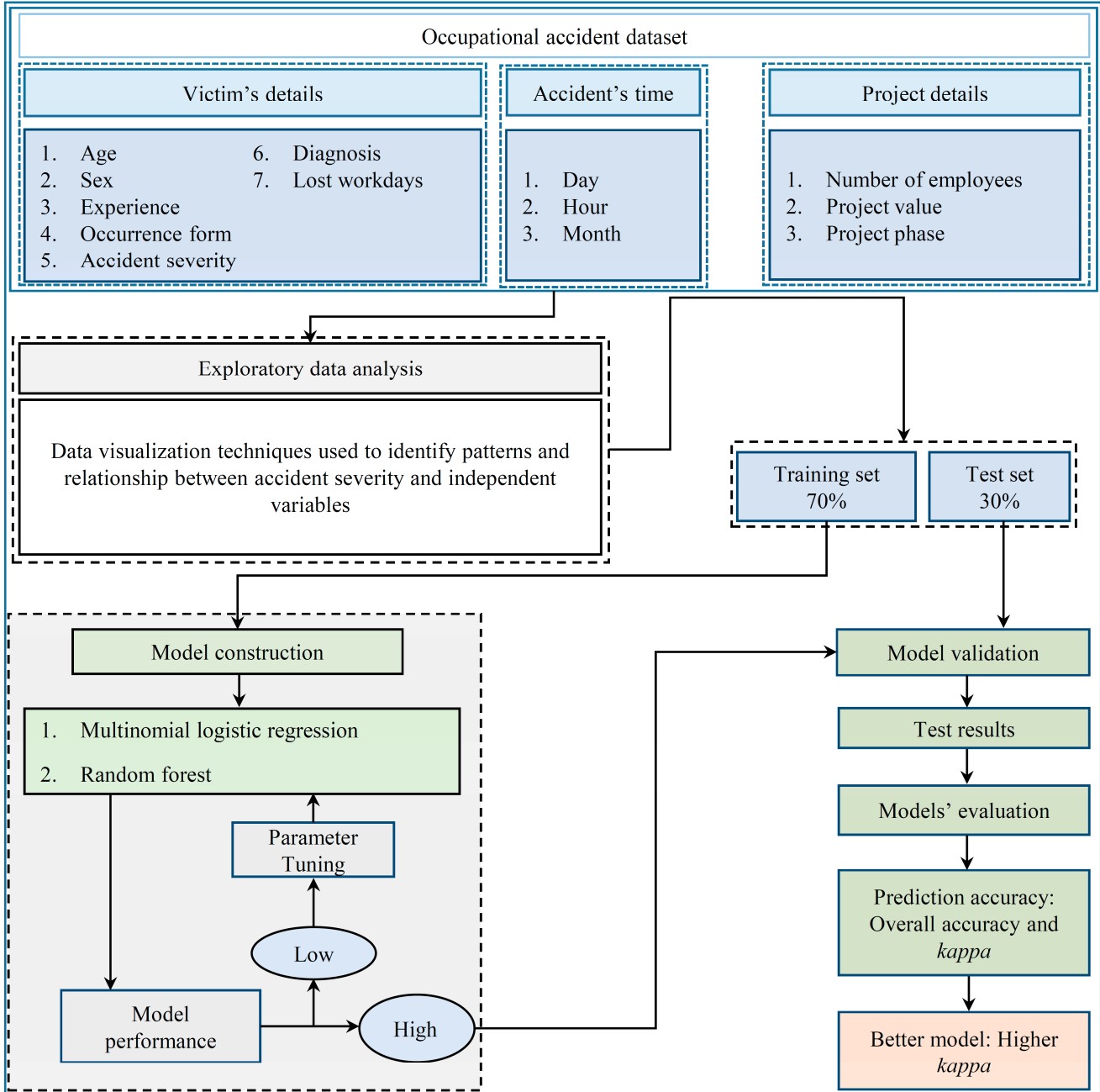

**Figure 1.** Occupational accident analysis workflow.

**Table 1.** Descriptive statistics of occupational accident data.

| | Categories | Accidents (in 100% of 27,211 Cases) | | | |
| --- | --- | --- | --- | --- | --- |
| | | Deaths | Disease Patients | Injuries | Total |
| Variables | % of victims | 2.19 | 6.7 | 91.11 | 100 |
| Sex | Female | 0.02 | 0.15 | 2.69 | 2.87 |
| | Male | 2.17 | 6.55 | 88.42 | 97.13 |
| Age (Years) | 18–24 | 0.01 | 0.01 | 0.51 | 0.53 |
| | 25–29 | 0.06 | 0.08 | 1.76 | 1.89 |
| | 30–34 | 0.05 | 0.09 | 2.34 | 2.48 |
| | 35–39 | 0.08 | 0.21 | 3.76 | 4.05 |
| | 40–44 | 0.14 | 0.32 | 5.81 | 6.27 |
| | 45–49 | 0.25 | 0.65 | 9.94 | 10.85 |
| | 50–54 | 0.37 | 0.89 | 14.99 | 16.26 |
| | 55–59 | 0.48 | 1.34 | 19.46 | 21.28 |
| | 60–64 | 0.35 | 1.72 | 18.65 | 20.72 |
| | 65–69 | 0.23 | 1 | 9.68 | 10.91 |
| | 70–74 | 0.12 | 0.28 | 3.43 | 3.83 |
| | 75–79 | 0.04 | 0.11 | 0.69 | 0.84 |
| | 80+ | 0 | 0.01 | 0.08 | 0.09 |
| Worker skills | Skilled | 1.12 | 3.34 | 43.11 | 47.58 |
| | Unskilled | 1.07 | 3.36 | 48 | 52.42 |
| Work experience | <1 month | 1.19 | 2.2 | 63.93 | 67.33 |
| | 1–2 months | 0.15 | 0.86 | 8.74 | 9.75 |
| | 2–3 months | 0.11 | 0.62 | 4.4 | 5.13 |
| | 3–6 months | 0.22 | 0.94 | 5.73 | 6.89 |
| | 6–12 months | 0.17 | 0.75 | 3.58 | 4.5 |
| | 1–3 years | 0.21 | 0.7 | 2.94 | 3.86 |
| | 3–5 years | 0.05 | 0.19 | 0.81 | 1.05 |
| | 5+ years | 0.1 | 0.43 | 0.96 | 1.48 |
| Diagnosis (injured part or infected organs of the victim) | Abdomen, lower back | 0.08 | 1.14 | 8.39 | 9.61 |
| | Ankle and foot | 0 | 0.01 | 11.84 | 11.86 |
| | Burns and corrosions | 0.05 | 0 | 1.37 | 1.43 |
| | Circulatory system | 0.23 | 0.32 | 0.03 | 0.59 |
| | Ear and mastoid process | 0 | 0.49 | 0.04 | 0.53 |
| | Elbow and forearm | 0 | 0.02 | 7.68 | 7.7 |
| | Head | 0.93 | 0.01 | 7.35 | 8.29 |
| | Hip and thigh | 0 | 0 | 2.79 | 2.79 |
| | Knee and lower leg | 0.01 | 0.13 | 11.86 | 12 |
| | Musculoskeletal system | 0.01 | 2.98 | 0.27 | 3.26 |
| | Neck | 0.1 | 0.03 | 1.54 | 1.66 |
| | Others | 0.47 | 0.26 | 1.57 | 2.3 |
| | Respiratory system | 0.09 | 0.57 | 0.04 | 0.7 |
| | Shoulder and upper arm | 0.01 | 0.67 | 4.37 | 5.05 |
| | Thorax | 0.21 | 0.06 | 9.56 | 9.83 |
| | Wrist and hand | 0 | 0.03 | 22.39 | 22.42 |
| Accident occurrence form | Building collapse | 0.09 | 0 | 1.23 | 1.32 |
| | Bump | 0.12 | 0 | 7.8 | 7.92 |
| | Carelessness | 0 | 0 | 2.81 | 2.81 |
| | Chemical contact | 0.03 | 0.01 | 0.24 | 0.27 |
| | Cutting and stabbing | 0.01 | 0 | 10.49 | 10.5 |
| | Disease infection | 0.32 | 6.68 | 0.02 | 7.03 |
| | Electric shock | 0.04 | 0 | 0.48 | 0.52 |
| | Entrapment | 0.08 | 0 | 7.62 | 7.7 |
| | Explosion and rupture | 0.02 | 0 | 0.18 | 0.2 |
| | Extreme heat exposure | 0 | 0 | 0.52 | 0.52 |
| | Fall down | 0.02 | 0 | 15.12 | 15.14 |
| | Fall from height | 1.1 | 0 | 30.38 | 31.48 |

**Table 1.** *Cont.*

| Categories | | Accidents (in 100% of 27,211 Cases) | | | |
| --- | --- | --- | --- | --- | --- |
| | | Deaths | Disease Patients | Injuries | Total |
| Accident occurrence form | Fire | 0.02 | 0 | 0.18 | 0.21 |
| | Flip over (buried) | 0.08 | 0 | 2.55 | 2.63 |
| | Hit by an object | 0.1 | 0 | 10.77 | 10.87 |
| | Off-site traffic accident | 0.1 | 0 | 0.56 | 0.65 |
| | Others | 0.06 | 0.01 | 0.16 | 0.22 |
| Project phase (% rate of completion) | 0–20 | 0.47 | 0.69 | 15.49 | 16.65 |
| | 20–40 | 0.38 | 1.01 | 18.31 | 19.7 |
| | 40–60 | 0.46 | 1.36 | 20.05 | 21.87 |
| | 60–80 | 0.43 | 1.62 | 19.97 | 22.02 |
| | 80+ | 0.45 | 2.02 | 17.29 | 19.76 |
| Number of workers | <5 | 0.76 | 1.9 | 37.76 | 40.42 |
| | 5–9 | 0.25 | 0.72 | 11.85 | 12.82 |
| | 10–15 | 0.18 | 0.49 | 8.15 | 8.83 |
| | 16–29 | 0.17 | 0.76 | 10.4 | 11.33 |
| | 30–99 | 0.36 | 1 | 11.88 | 13.25 |
| | 100–499 | 0.36 | 1.34 | 8.54 | 10.24 |
| | 500+ | 0.1 | 0.49 | 2.52 | 3.11 |
| Project value (in million KRW) | 10–20 | 0.32 | 0.52 | 11.87 | 12.71 |
| | 20–40 | 0.17 | 0.38 | 6.22 | 6.77 |
| | 40–100 | 0.16 | 0.4 | 8.28 | 8.84 |
| | 100–500 | 0.36 | 1 | 18.06 | 19.42 |
| | 500–1000 | 0.18 | 0.44 | 7.95 | 8.57 |
| | 1000–5000 | 0.32 | 1.16 | 16.85 | 18.33 |
| | 5000–10,000 | 0.12 | 0.47 | 6.1 | 6.69 |
| | 10,000–50,000 | 0.22 | 0.67 | 6.12 | 7 |
| | 50,000+ | 0.35 | 1.67 | 9.65 | 11.67 |
| Month of accident | January | 0.15 | 0.57 | 6 | 6.73 |
| | February | 0.12 | 0.42 | 4.68 | 5.22 |
| | March | 0.21 | 0.58 | 7.31 | 8.1 |
| | April | 0.15 | 0.55 | 7.76 | 8.46 |
| | May | 0.18 | 0.63 | 8.59 | 9.4 |
| | June | 0.2 | 0.62 | 7.77 | 8.58 |
| | July | 0.24 | 0.64 | 8.31 | 9.19 |
| | August | 0.22 | 0.6 | 8.5 | 9.33 |
| | September | 0.17 | 0.36 | 6.41 | 6.93 |
| | October | 0.12 | 0.59 | 9.2 | 9.92 |
| | November | 0.22 | 0.57 | 9.01 | 9.79 |
| | December | 0.21 | 0.56 | 7.58 | 8.35 |
| Day of accident | Sunday | 0.17 | 0.2 | 5.73 | 6.1 |
| | Monday | 0.37 | 1.4 | 14.51 | 16.28 |
| | Tuesday | 0.37 | 1.19 | 14.93 | 16.49 |
| | Wednesday | 0.38 | 1.13 | 14.47 | 15.98 |
| | Thursday | 0.28 | 1.11 | 14.09 | 15.47 |
| | Friday | 0.35 | 1.1 | 14.88 | 16.33 |
| | Saturday | 0.28 | 0.57 | 12.51 | 13.35 |
| Hour of accident | 0:00 | 0.09 | 1.71 | 1.18 | 2.98 |
| | 1:00 | 0.01 | 0.01 | 0.17 | 0.19 |
| | 2:00 | 0 | 0.01 | 0.16 | 0.17 |
| | 3:00 | 0.01 | 0 | 0.12 | 0.13 |
| | 4:00 | 0 | 0 | 0.05 | 0.05 |
| | 5:00 | 0.01 | 0.02 | 0.1 | 0.13 |
| | 6:00 | 0.04 | 0.03 | 0.32 | 0.4 |
| | 7:00 | 0.1 | 0.1 | 3.05 | 3.25 |
| | 8:00 | 0.23 | 0.39 | 8.2 | 8.82 |

**Table 1.** *Cont.*

| Categories | | Accidents (in 100% of 27,211 Cases) | | | |
| --- | --- | --- | --- | --- | --- |
| | | Deaths | Disease Patients | Injuries | Total |
| Hour of accident | 9:00 | 0.2 | 1.35 | 10.07 | 11.63 |
| | 10:00 | 0.29 | 0.94 | 14.11 | 15.34 |
| | 11:00 | 0.2 | 0.4 | 10.58 | 11.18 |
| | 12:00 | 0.08 | 0.22 | 2.31 | 2.61 |
| | 13:00 | 0.21 | 0.19 | 7.11 | 7.52 |
| | 14:00 | 0.22 | 0.44 | 11.39 | 12.04 |
| | 15:00 | 0.17 | 0.44 | 10.82 | 11.43 |
| | 16:00 | 0.17 | 0.26 | 8.36 | 8.79 |
| | 17:00 | 0.07 | 0.07 | 1.84 | 1.99 |
| | 18:00 | 0.03 | 0.05 | 0.47 | 0.55 |
| | 19:00 | 0.02 | 0.02 | 0.23 | 0.27 |
| | 20:00 | 0.01 | 0.02 | 0.15 | 0.18 |
| | 21:00 | 0.01 | 0.01 | 0.12 | 0.14 |
| | 22:00 | 0.01 | 0.01 | 0.1 | 0.12 |
| | 23:00 | 0.01 | 0 | 0.08 | 0.1 |
| Lost workdays | Discrete variable | 11.00 | 12.00 | 77.00 | 100 |

### 2.2. Methods

This section discusses the detailed methodologies used to perform all analyses, including the exploratory data analysis techniques and the modeling method with their assessment techniques.

### 2.2.1. Exploratory Data Analysis Techniques

Understanding the patterns and visualization of clusters in the dataset is an important step before going further in the modeling method. This method helps to understand data distribution and the identification of issues (anomalies), such as missing data or outliers, within the dataset through the graphical representation of data. To accomplish this, the following techniques were applied:

- Cleveland dot: The Cleveland dot plot is better at reducing the visual cutter in the plot and makes it easier to view patterns in the data on the graph [30].
- Balloon plot: It graphically represents the contingency (matrix) table with a size dot corresponding to the entry value in the matrix in the categorical data. The balloon plot serves as an analogy for the correlation matrix when analyzing correlations in the context of continuous variables [31].
- Correspondence analysis: It facilitates the visualization of cross-tabulated data that are on the same scale while considering the weight of factors. It enables a visual representation that takes into account the relative importance or weight of the factors being analyzed [32]. The category with more weight tends to be in the center of the graph, while the categories with few counts tend to be far from the center. Correlated categories are located on the same side of the hyperplane, while negatively correlated categories are placed on opposite sides of the axis.

### 2.2.2. Modeling Techniques

Multinomial logistic regression is widely used to model multi-categorical outcome variables where the dependent variable is nominal, with an unordered outcome. In this model, logarithms of odds (known as logit) of the dependent variable are expressed in the

form of linear combinations of independent variables [33]. The general equation of the logarithm of odds is written as follows,

$$\log(odds) = logit(p) = \ln\left(\frac{p}{1-p}\right) = a_0 + a_1\ x_1 + a_2\ x_2 + \cdots + a_n\ x_n$$
$$\text{with} \tag{1}$$
$$p = \frac{\exp(a_0 + a_1\ x_1 + a_2\ x_2 + \cdots + a_n\ x_n)}{1 - \exp(a_0 + a_1\ x_1 + a_2\ x_2 + \cdots + a_n\ x_n)}$$

where $p$ is the probability that a given outcome falls in a particular category, exp() is the exponential function with base (e $\approx$ 2.72), and $a_i$ are coefficients of independent variables. In order to assess the goodness of fit of the multinomial model and remove non-contributing variables, the Z-test was conducted. To conduct a comparative evaluation of the multinomial model's performance, the random forest model was chosen for its high accuracy in multi-classification problems [34]. This comparative study is crucial to identify the difference in performance so that alternative methods may be adopted when the accuracy of the proposed method is not satisfactory.

## 3. Results and Discussion

### 3.1. Results

#### 3.1.1. Exploratory Data Analysis

Exploratory data analysis exhibits the relationship between the severity of the accident and the independent variables. Overall, 27,211 accident cases occurred in the year 2019, with 91% of them being bodily injuries, 7% were occupational diseases, and 2% were fatal accidents (Table 1). Observations revealed that the gender ratio for all accidents was 97% males and 3% females.

Different graphical techniques were explored to graphically visualize the interconnection between accident severity and influencing factors. The correspondence analysis was used to summarize the relationship between categorical variables in a contingency table. From the contingency table, the calculation of the proportion of inertia associated with rows and columns based on their profile represents the singular value decomposition (factorization of matrix diagonal with positive real number entries and unitary matrix) in the two-dimensional plot [35]. The balloon plot represents a graphical matrix that contains dots whose size reflects the magnitude of the associated component [31]. Figure 2a depicts that accident frequency was higher for workers with less work experience, especially below one month, and 98% of the variability in the data can be explained with the first dimension. This association of accidents is also reflected in the balloon plot (Figure 2b), where a large balloon represents the correspondence between injuries and individuals with less than one month of experience. The bigger the balloon, the closer to the center of the graph, and the fewer cases, the longer the distance from the center. Disease patients were more likely to have more work experience, which is explained by an acute angle [36], between three and five years of work experience, while deaths were more likely associated with more than one year of employment.

The impact of work experience on the frequency and severity was deeply analyzed based on the form of occurrence of the accident. Figure 3 shows that falling from height was the most frequent among workers with less than one year of experience, with a frequency of 6235 (22.9%) of all accident cases, followed by falling down on the same level with a frequency of 2782 (10.22% of all accident cases). In total, 67% of all accident victims were new workers with less than one month on the job. As the employment time increased, the frequency of falling from height accidents fell sharply from 22.9% to 2.7% of all cases. It was observed that the initial sources of the accidents were falling from a height, falling down, hit by an object, cutting and stabbing, bumped into an object, and entrapment, which totaled 83.62% (22,753) of all accidents.

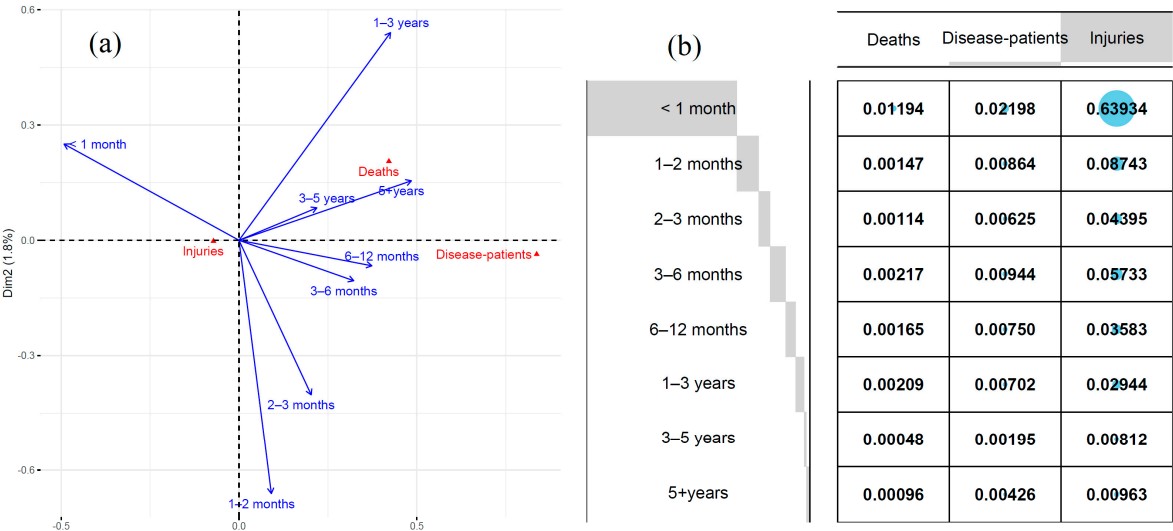

**Figure 2.** (**a**) Biplot represents the relationship between employment period and severity of accidents, and (**b**) balloon plot represents proportions of accidents and their association with work experience.

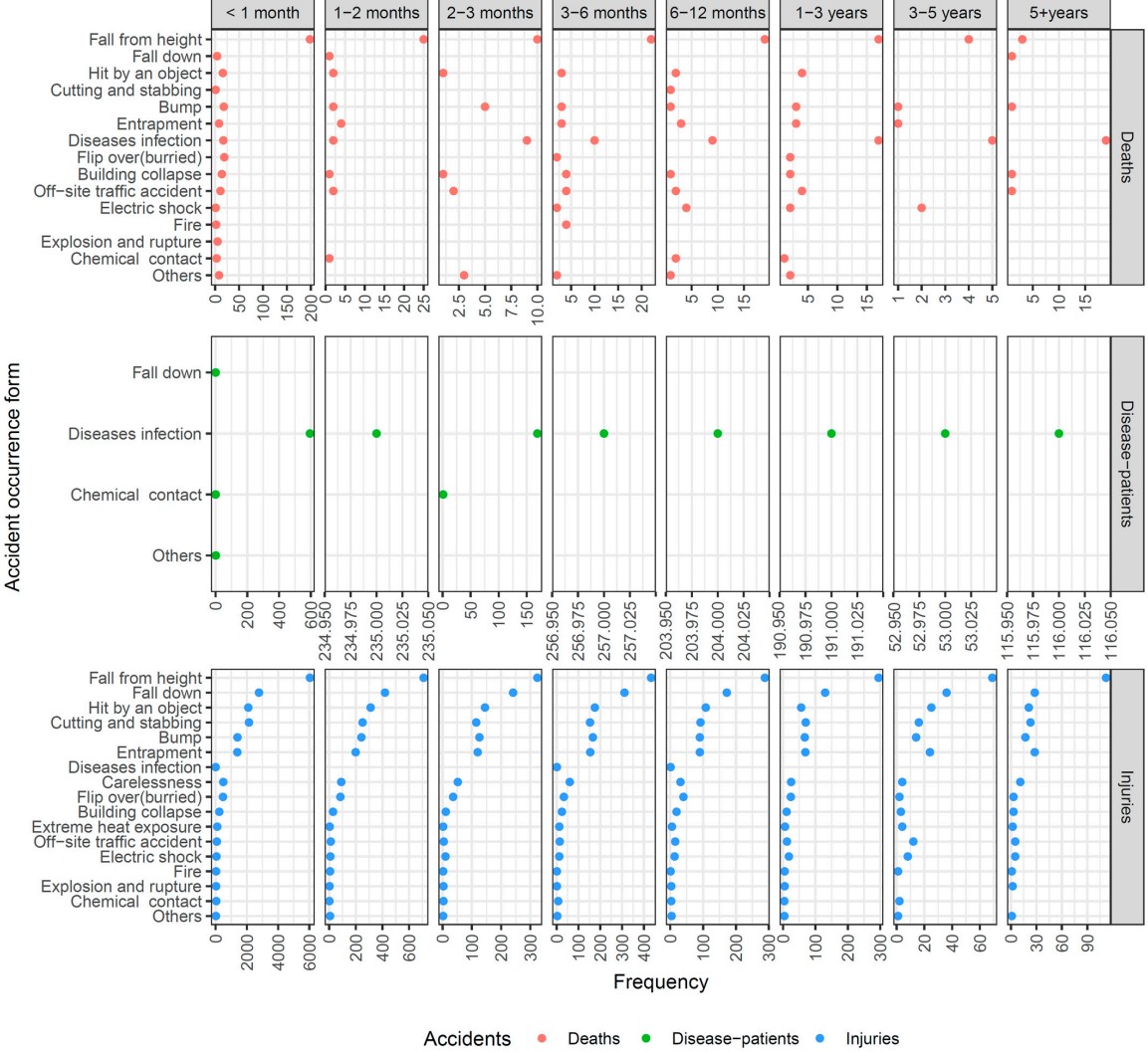

**Figure 3.** Distribution of accidents by the form of occurrences and work experience.

When an accidental incident occurred, the victim stopped working for medical care once the event did not swipe away his life. The loss of workdays may be influenced by the severity of the accident and other different factors such as age, gender of the victim [37], work experience, etc. Figure 4 shows that the loss of workdays for fatal accidents was above 1000 days, and the workers with less than one month of work experience had a higher mode (325), totaling 54.5% of all fatal accidents. The mode of fatal accidents shifted to 40 (7% of all fatal cases) among workers with one to two months of work experience, highlighting the heightened vulnerability of new workers, and subsequent work experience intervals maintained a lower rate of fatal accidents. The decline in accidents after the first month could be attributed to increased adaptability to working conditions.

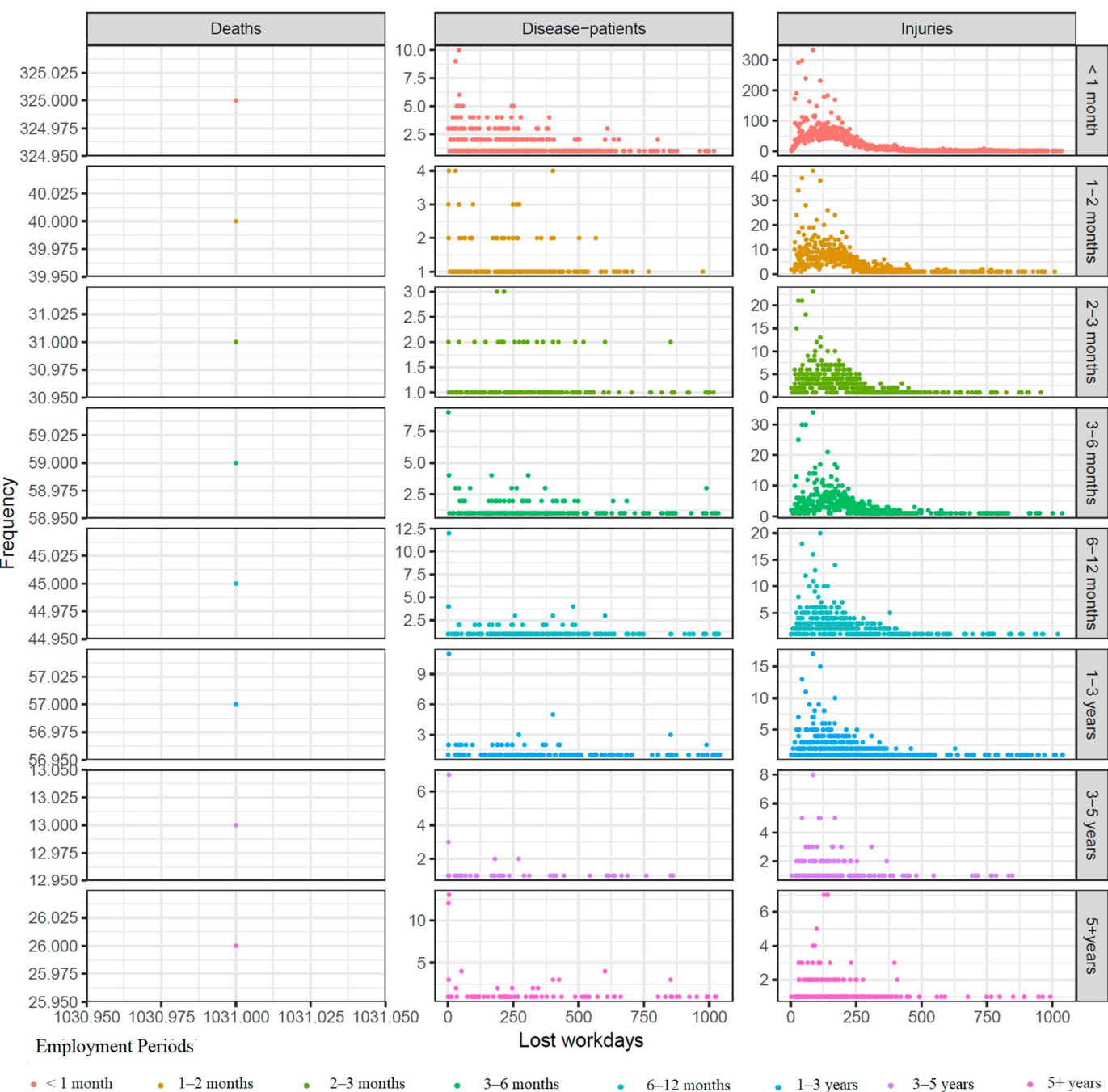

**Figure 4.** Distribution of lost workdays by work experience and accident severity.

The frequency of the loss of working days remained low and did not show observable differences based on work experience among disease patients. However, for injuries, the most common duration of lost workdays among novice workers was 84 days with a frequency of 332 times, followed by 42 days with a frequency of 297 times. Similarly, for workers with one to two months of work experience, the most frequent (mode) durations of lost workdays were still 84 days and 42 days, with frequencies of 42 and 39, respectively.

Due to the change in human homeostasis and energy over the day, the hour of the day is believed to have an impact on the occurrence of the accident [11]. Figure 5 shows that there is a difference in the hour of the accidents. It is observable that workers became contaminated with diseases mostly between 9:00 and 10:00 a.m. The higher frequency of contamination occurring around midnight (about 500 cases) may be due to erroneous reporting of nightshift accidents, which may be allocated by default when the hour was not correctly recorded or because of the tiredness associated with sleep loss caused by circadian disturbance. Fatal accidents exhibited two peaks, one at 10:00 a.m. and the second at 2:00 p.m, the same as accidental injuries. A noticeable decrease in the number of accident cases is observed at 12:00 p.m. and 1:00 p.m., which can be attributed to lunchtime breaks. It is important to note that Figure 5 was plotted on different scales to enhance the visualization of patterns in the data, not to indicate the weight or magnitude of the cases. Furthermore, there was no discernible variation in the pattern of accidents per hour across different days. The two peaks at 10:00 a.m. and 2:00 p.m. remained consistent throughout the week, with a lower frequency observed on Sundays.

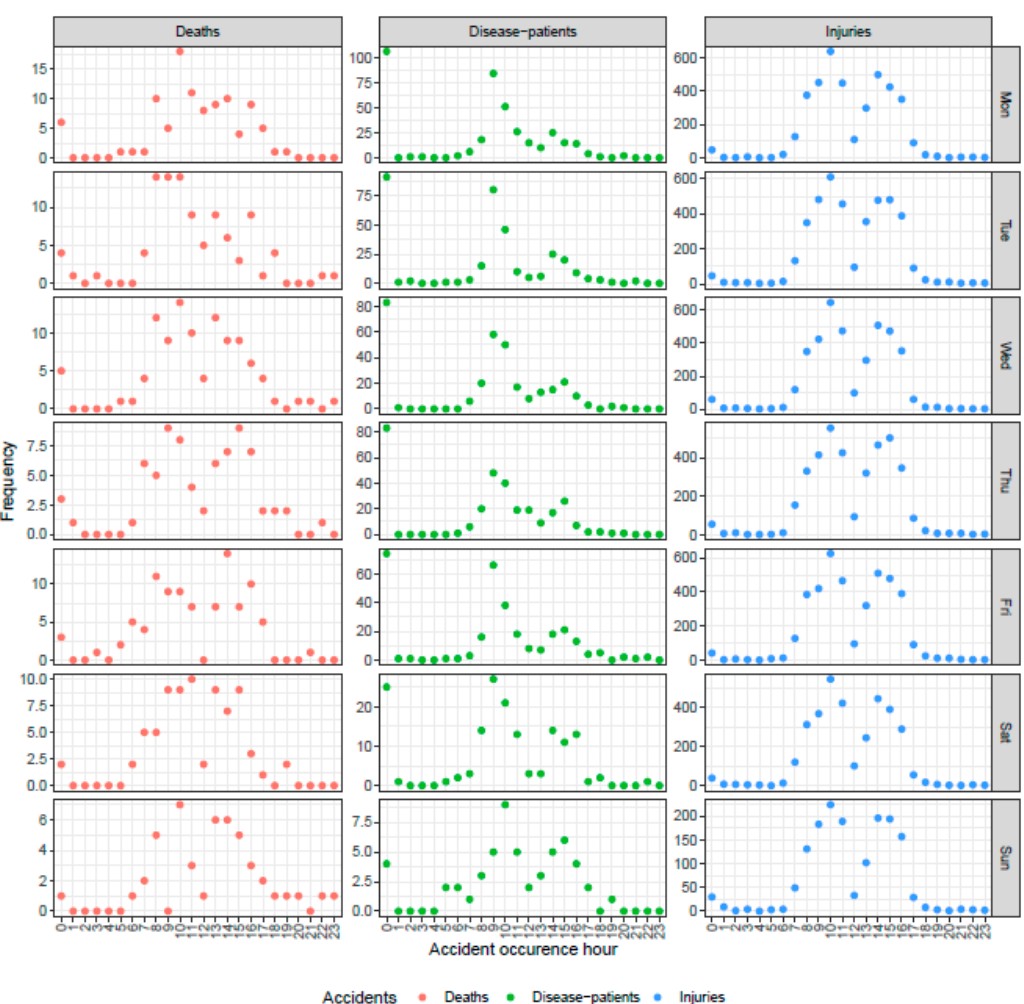

**Figure 5.** Distribution of accidents by hour of occurrence.

Figure 6 shows the distribution of occupational accidents based on the gender and age of the victims at the time of the accident. A significant disparity is observed between males and females in terms of accident frequency. The pattern of accidents is similar among skilled and unskilled workers, with a higher frequency occurring among workers between the ages of 50 and 64, regardless of gender. The higher frequency of accidents among individuals aged 50 to 64 does not necessarily imply that they are more vulnerable. Instead, it reflects the age distribution of the Korean population as represented by the population pyramid.

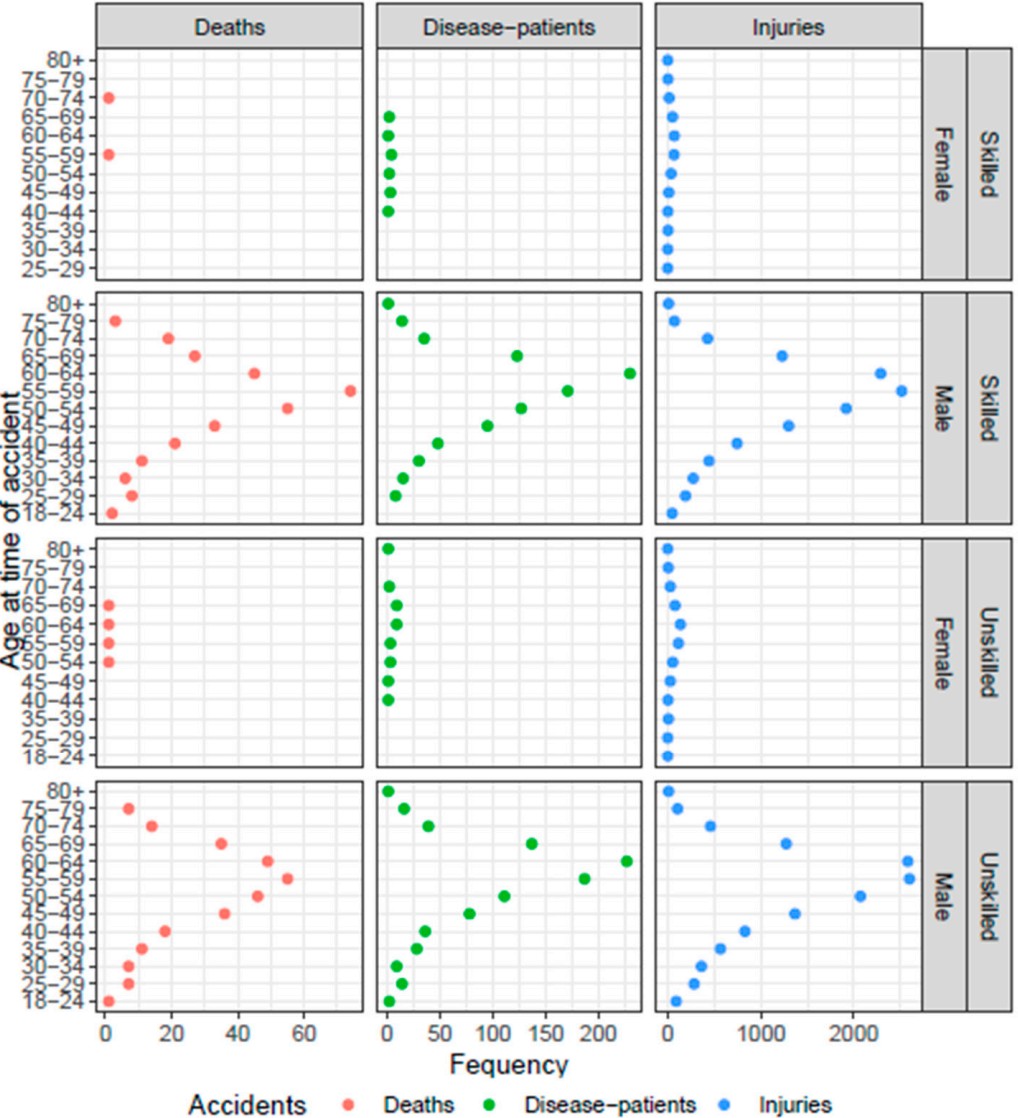

**Figure 6.** Age- and gender-based distribution of accidents by worker's skills.

The body part injured may play a tremendous role in the identification of the severity of the accident. This is because certain body parts may require a longer time to heal, while injuries to other parts may pose a higher risk of fatality. Figure 7 shows that the most frequently injured parts, namely, the wrist and hands and the knee and lower leg, did not lead to fatal accidents, while neck, thorax, head, abdomen, and lower back injuries were likely to lead to fatal accidents. In order to facilitate clear visualization of the data, Figure 7 was plotted on different scales due to the rapid decrease in frequency from the first month to the 1–3 months' interval and beyond.

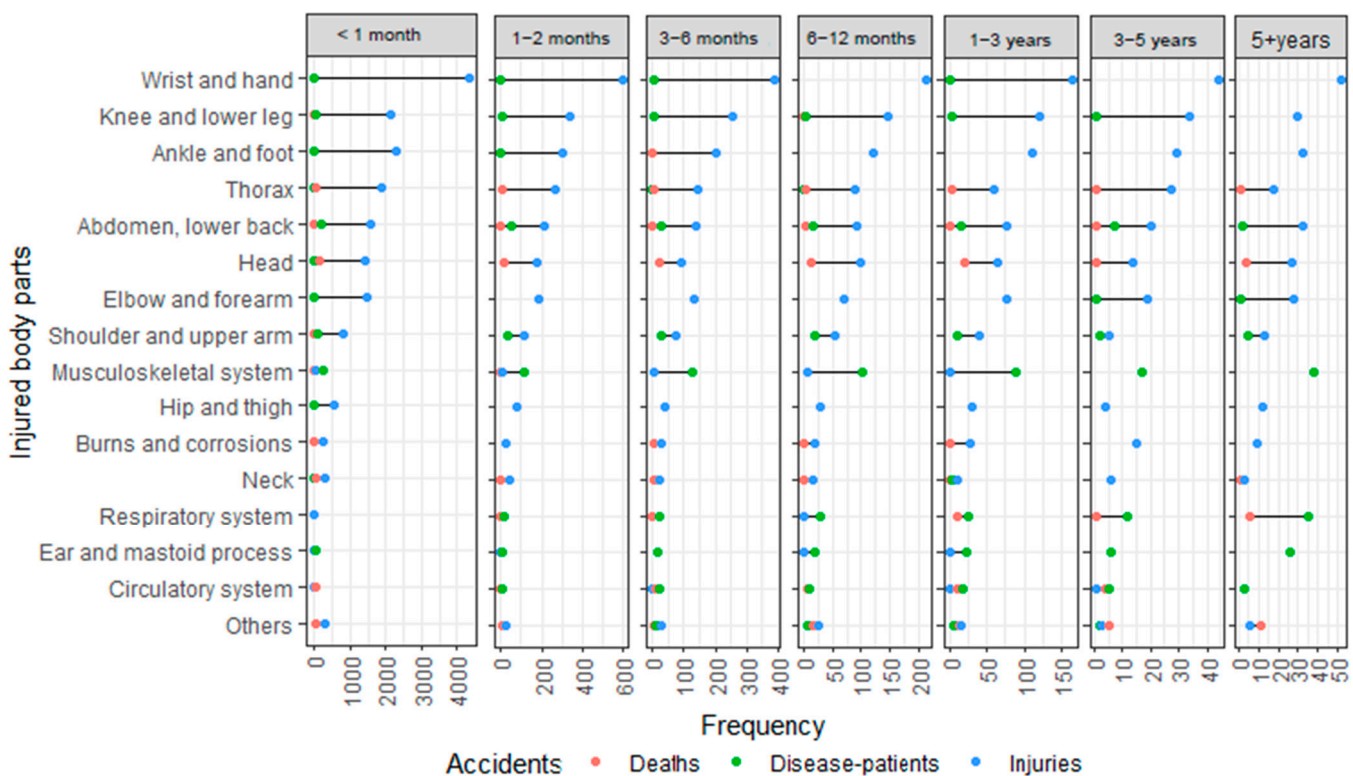

**Figure 7.** The severity of accidents based on the injured part of the body and work experience.

### 3.1.2. Accident Severity Prediction and Model Validation

The multinomial logistic regression model was used to estimate the relationship between the severity of the accident and independent variables. In order to utilize this model, the null hypothesis stated that all selected predictor variable coefficients are zero, indicating that those variables have no influence on the severity of accidents. The alternative hypothesis proposed that some predictor variables considered for the multinomial logistic regression have non-zero coefficients, indicating their influence on the severity of accidents. To estimate the impact of predictor variables, the null model (the model with only a constant) was constructed and compared with the full model (model with all variables). Following the construction of the null and full models, a likelihood ratio test was conducted to compare the two models and determine whether they have a significant difference. This test is commonly used to assess the goodness of fit of a multinomial model [38]. In order to select the optimal model and remove non-influential variables from the full model, a two-tailed Z-test was conducted for each presumed influential variable. A variable with *p*-values greater than 0.05 for all categorical levels was considered non-significant and subsequently removed from the full model [39]. The full model included variables such as age, sex, workers' skills, work experience, completion rate of the project or working phase, number of employees, project value, diagnosis (indicated by the International Classification of Disease Code, IDC-11 [28]), and the form of occurrence of an accident (direct source of accident). After conducting the two-tailed test, it was determined that age, workers' skills, project value, and day of the accident were insignificant. As a result, they were removed from the full model to obtain the optimal model.

To quantify the proportion explained by the predictor variable in the model, pseudo R-squared values were calculated. Pseudo R-squared measures the proportion of variance in the dependent variable, which was explained by the explanatory variables. CoxSnell, Nagelkerke, and McFadden R-squared were used to explain differences between the null, full, and optimal models [40]. CoxSnell R-squared is the analogy of R-squared in the ordinary least-square regression but with a different meaning. CoxSnell's minimum is zero, and its maximum is less than one, which makes it easy to explain. The modified

version, Nagelkerke R-squared, ranges between zero and one, making it easier to interpret and a more reliable measure of the relationship between variables. The McFadden R-squared is the ratio of the difference in log-likelihood between the full model and the null model, divided by the log-likelihood of the null model [40–42]. It provides a measure of the improvement in model fit, with higher values indicating a better fit. The McFadden R-squared is calculated using the following equation,

$$\text{McFadden} = \frac{\text{LL(Null model)} - \text{LL(Full or optimal model)}}{\text{LL(Null model)}} \qquad (2)$$

where LL stands for log-likelihood.

To select the optimal model, the performance of the training set and the complexity of the model were assessed using the Akaike information criterion (*AIC*) and Bayesian information criterion (*BIC*). In comparing multiple models with different variables, the model with a lower value of *AIC* and *BIC* and higher performance is considered optimal. That is the model with fewer predictors than the full model that performs the same or better than the full model. Hastie et al. [43] defined *AIC* and *BIC* as

$$AIC = -\frac{2}{N} * LL + \frac{2K}{N} , \quad and \quad BIC = -2 * LL + \ln(N) * K \qquad (3)$$

where $N$ represents the number of observations in the training set, $K$ is the number of independent variables in the model, and ln is the natural logarithm.

The results for pseudo R-squared (Table 2) show that there is a difference between the null and the full model. CoxSnell's R-squared was 0.441, which means that predictor variables explained 44.1% of the severity of accidents. On the other hand, Nagelkerke and McFadden show that 87.75% and 83.28% of the variance in the severity of accidents was explained by the predictor variable.

**Table 2.** Pseudo R-squared for the multinomial models.

| | Pseudo R-Squared | | | Information Criterion | |
|---|---|---|---|---|---|
| **Models** | **CoxSnell** | **Nagelkerke** | **McFadden** | **AIC** | **BIC** |
| Null model | 0 | 0 | 0 | 13,325.970 | 13,341.670 |
| Full model | 0.441 | 0.877 | 0.832 | 2670.543 | 4414.290 |
| Optimal model | 0.441 | 0.877 | 0.833 | 2559.036 | 3878.629 |

*3.2. Discussion*

The proposed optimal model excluded age, worker skills, project value, and the day of accident predictors, which did not contribute to the performance of the model. The significant variables, their odds ratio, confidence interval, and corresponding *p*-values are summarized in Table S1 (Supplementary Data). The odds ratio is the ratio of the probability of being in a given category and the probability of not being in that category (probability of success divided by the probability of failure). The category death was considered as the reference category of the multinomial model. Some variables demonstrated greater influence on injuries, whereas others exhibited heightened significance in relation to the occurrence of injuries. For example, work experience exhibited noteworthy influence as a variable in relation to occupational injuries, but it did not yield the same significance in the context of occupational diseases.

Overall, most variables were not significant for risks of diseases because the risk was mostly from work itself, not the working environment. The risk ratio (odds ratio) of diseases versus death slightly decreased from June to September and in December. The worker's gender significantly differentiates fatal accidents from disease patients and injuries from fatal accidents. Specifically, an accident that happened to males multiplies the odds of being injured versus fatal accidents by 0.6 (*p* < 0.05). Work experience significantly distinguished the occurrence of injuries versus fatal accidents in the 3–6 months, 6–12 months, and

1–3 years of employment periods versus workers with less than one month of experience, with odds of being injured of 0.47 ($p$ = 0.002), 0.51 ($p$ = 0.008), and 0.39 ($p$ < 0.001) (for respective experience periods) versus fatal accidents. This reduction in injuries may be due to adaption to working conditions, and the increase in the rate of fatal accidents may be attributed to the assignment of more dangerous tasks as the work experience increases.

The completion rate (project phase), at all levels, had odds of an accident being an injury versus fatal between 60% and 102%. For the case of the number of employees, workplaces with fewer workers had a higher likelihood of fatal accidents. The employment place with 5–9 workers had 44% (1–0.64) less odds of injuries vs. deaths than places with 0–5 workers. For the accident hour, the hour of the accident was not a significant determinant of the severity of the accident except at 6:00 a.m. and 7:00 p.m. where odds of injuries versus deaths were 82% (0.18–1) and 89% (0.11–1) less, respectively, with respect to 0:00 a.m. (the reference hour). This reduction in the injury ratio may have originated from those two hours being a transition period from dayshift to nightshift working time.

For diagnosis cases with the reference category being the abdomen and lower back, burn and corrosion injuries to the head and neck had a higher likelihood of causing death over injuries compared to the abdomen and lower back accidents. The form of the occurrence variable, cutting and stabbing, falling down, hit by an object, and bumps, had 52.2, 22.5, 9.97, and 5.23 more odds of injuries over death compared to the reference level (building collapse), respectively. On the contrary, the disease infection had 0.07 less odds of injuries over death with respect to the building collapse category.

To verify the performance of the multinomial model, it was compared to the random forest model, as illustrated in Table 3. It was observed that both models exhibited excellent performance, with an accuracy of 97.66% and 97.7%, respectively, on the test dataset. Despite the high performance of 0.04% for the random forest over the multinomial model, the multinomial model had higher overall predictive power, as indicated by its higher kappa value.

**Table 3.** Comparing multinomial model and random forest performance.

| Values | Multinomial | Random Forest |
|---|---|---|
| Accuracy | 0.9766 | 0.9738 |
| 95% CI | (0.9731, 0.9798) | (0.9735, 0.9801) |
| No Information Rate | 0.9111 | 0.9111 |
| $p$-Value [Acc > NIR] | <0.001 | <0.001 |
| Kappa | 0.8463 | 0.8443 |

Furthermore, the receiver operating characteristic (ROC) [44,45] curve exhibits that both multinomial and random forest models performed better by accurately classifying 100% of the victims who contracted diseases (Figure 8). In other cases, the multinomial model classified injuries and fatalities far better than the random forest model. The fatal cases were mostly misclassified in the random forest model. The classification accuracy for the injuries category was still far higher in the multinomial than in the random forest model. Based on these findings, the multinomial model can be used as a predictive model for the severity of occupational accidents due to its superiority in ranking accidents in their appropriate categories.

The finding in this research study represents the situation of occupational accidents for the case study of South Korea for the accident records of 2019. The results revealed that 67% of all accidents occurred within the first month of the employment period, and the prominent accident was falling from a height, while Jeong [2] found that about 50% of all accidents that occurred between 1991 and 1994 happened within the first year and the main accident cause was entrapment. However, in the present study, entrapment ranked as the sixth cause of accidents (Figure 3), which shows a shift in the cause of accidents over time. In addition, it was observed that the most frequently injured parts, namely, wrist, hand, knee, and lower leg, did not lead to fatal accidents, while the neck, thorax, head,

abdomen, and lower back were more often sources of fatal accidents. This accident pattern has similarities with the observation of Al-Abdallat et al. [10], where head injuries led to fatalities and permanent disability in Jordan.

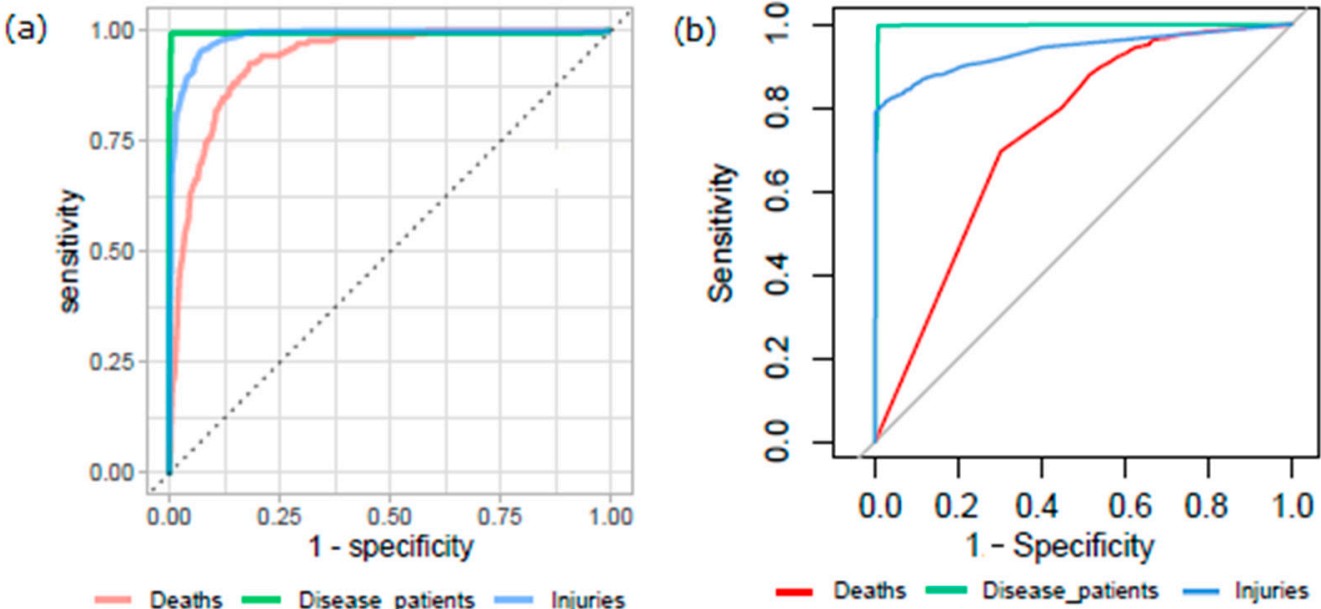

**Figure 8.** ROC curve of the (**a**) multinomial and (**b**) random forest model.

Further studies are needed to understand occupational accident patterns in general. In the considered data, the categorization of workers into skilled or unskilled groups was taken into consideration; however, their educational levels and vocational training remained undisclosed. Furthermore, it is essential to note that the timing of accidents reflected the moment of occurrence rather than the cumulative work hours leading up to the accidents. Given that fatigue has the potential to impact accident likelihood, as highlighted by Swaen et al. [46], there is a compelling need to enhance the database by introducing supplementary variables that are believed to influence the occurrence of occupational accidents. In addition, collecting geo-localization information would be helpful for future studies in identifying clusters of accidents in different regions within the country. Other datasets, such as environmental climate variables, safety management, and safety technology, could also play a pivotal role in strengthening safety protocols in regions with higher susceptibility to accidents.

## 4. Conclusions

This study aimed to assess the severity of occupational accidents in South Korea by considering various influencing factors. The initial phase involved exploratory data analysis to explore the connection between accident severity and independent variables visually. Subsequently, a predictive model utilizing multinomial logistic regression was developed to anticipate the condition of accident victims. To assess the model's accuracy, a comparison was made against a random forest model. The outcome revealed that 67% of all accident victims were newly recruited employees with less than one month of experience. Furthermore, accidents were most frequent at 10:00 am, constituting 15% of all cases, closely followed by 2:00 pm, accounting for 12% of all incidents. The predictive accuracy of the multinomial regression model was determined to be 97.66% on the test dataset, with a kappa of 0.846. This performance surpassed that of the random forest model across all categories. Both models perfectly predicted the disease patient category; however, the random forest model was less accurate in predicting fatal accidents and injuries.

Given the heightened accident rate within the first month of employment, several strategies are recommended to reduce the frequency of occupational accidents. These

strategies include providing comprehensive safety education and limiting the exposure of new recruits to the most hazardous work areas until they have gained sufficient experience. The findings of this study offer valuable insights for shaping appropriate worker protection policies. Such policies could encompass ongoing monitoring, tailored safety education based on job roles, emphasis on recognized hazards, and setting insurance reserves to assist victims in post-accident relief.

**Supplementary Materials:** The following supporting information can be downloaded at https://www.mdpi.com/article/10.3390/su152015058/s1, Table S1: Multinomial model odds ratios and *p*-values.

**Author Contributions:** Conceptualization, J.T., S.-G.Y., M.D.A. and T.-K.O.; Methodology, J.T. and T.-K.O.; Investigation, J.T., S.-G.Y. and T.-K.O.; Formal analysis, J.T., S.-G.Y., M.D.A. and T.-K.O.; Resources, T.-K.O. and S.-G.Y.; Writing—original draft preparation, J.T., M.D.A. and S.-G.Y.; Writing—review and editing, J.T., S.-G.Y., M.D.A. and T.-K.O.; Visualization, J.T. and M.D.A.; Supervision, S.-G.Y. and T.-K.O.; Project administration, S.-G.Y. and T.-K.O.; Funding acquisition, T.-K.O. All authors have read and agreed to the published version of the manuscript.

**Funding:** This research was supported by the Basic Science Research Program through the National Research Foundation of Korea (NRF) funded by the Ministry of Education (No.2021R1I1A2050912).

**Institutional Review Board Statement:** Not applicable.

**Informed Consent Statement:** Not applicable.

**Data Availability Statement:** The datasets used and/or analyzed during the current study are available from the corresponding author on reasonable request.

**Acknowledgments:** The authors highly appreciate the anonymous reviewers and the editors for their constructive comments and suggestions that helped us to improve the paper.

**Conflicts of Interest:** The authors declare no conflict of interest.

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
