# Peer review of "Analysis of the Severity and Cause and Effect of Occupational Accidents in South Korea"

_sustainability, doi:10.3390/su152015058_

Round 1

Reviewer 1 Report

The text of the article should be carefully reviewed as there are numerous linguistic errors and typos, as well as other mistakes, for example:

line 214: there should be Figure 2(a) instead of 1(a),

lines 166-178: different font size and type,

references contain the same literature items (17 and 18, 36 and 37).

Table 4 is hard to read. For better readability, the significant variables should be highlighted. It would also be useful to summarize the information in the table in a few sentences, indicating the variables that are significant.

The Authors should primarily cite the recent publications (meanwhile 20 percent of the items in the references were published before year 2000).

The article was written in hardly academic language. Furthermore, there are linguistic errors, typos, double spaces or missing spaces, etc. in the text.

Reviewer 2 Report

Authors want to analyse the severity and cause-effect of occupational accidents in South Korea. The whole paper is generally written, including the structure, contents and language. At the moment, I would not recommend the work for publication. However, following suggestions are given for improving this manuscript:

(1)it is incomplete in occupational accident analysis workflow, such as education, vocational training, environmental climate, safety management and safety technology.

(2)Can the authors develop software for automatic statistical analysis?

(3)it is a lack of innovation in the setting of methodology and the selection of working conditions, and the occupational accident analysis workflow is obviously insufficient. Therefore, the work content of this paper is not enough to support a complete academic paper.

(4)The presented data and their interpretation does not allow a positive assessment of this work.

Author Response

Please see the attached response letter.

Reviewer 3 Report

The article discusses an important topic concerning the analysis of accidents at work. The authors analyzed 27211 accidents that occurred in 2019 in South Korea. The results of the study can be an indication of what preventive measures can be taken to reduce the number of accidents. Nevertheless, I have a few comments.

Figure 2a not clear, figure 2b - the size of the dots is not quite clear. There are small differences between the size of the dots for death and disease patients, for work experience of 2 to 5 years.

Figure 3 - points sometimes overlap and not all dots can be seen exactly. Maybe reduce the diameter of the dots, or show the distribution of accidents in three separate drawings.

Figure 4,6 - dots merge - too large dot diameters.

Author Response

(The authors gave the same response as above.)
